Reversine, a selective MPS1 inhibitor, induced autophagic cell death via diminished glucose uptake and ATP production in cholangiocarcinoma cells

Prajumwongs Piya 1 2
Waenphimai Orawan 1 2
Vaeteewoottacharn Kulthida 1 2
Wongkham Sopit 1 2
Sawanyawisuth Kanlayanee kanlayanee@kkumail.com 1 2
1 Department of Biochemistry, Faculty of Medicine, Khon Kaen University , Khon Kaen , Thailand
2 Cholangiocarcinoma Research Institute, Faculty of Medicine, Khon Kaen University , Khon Kaen , Thailand
Eskelinen Eeva-Liisa
Electronic publication date: 2021 Jan 7
Publication date: 2021
Volume: 9
Electronic Location ID: e10637
Received 2020 Jun 8; Accepted 2020 Dec 2
Copyright: ©2021 Prajumwongs et al.
Copyright year: 2021
Copyright holder: Prajumwongs et al.
License: This is an open access article distributed under the terms of the Creative Commons Attribution License, which permits unrestricted use, distribution, reproduction and adaptation in any medium and for any purpose provided that it is properly attributed. For attribution, the original author(s), title, publication source (PeerJ) and either DOI or URL of the article must be cited.
License URL: https://creativecommons.org/licenses/by/4.0/

Keywords: Reversine, MPS1, Cholangiocarcinoma, Autophagy, Apoptosis, GLUT1

Funding: Khon Kaen University 6200020002 Faculty of Medicine, Khon Kaen University IN62110 Postgraduate Scholarship of Faculty of Medicine, Khon Kaen University This study was co-supported by the research grant from Khon Kaen University (6200020002), and the invitation research grant from the Faculty of Medicine, Khon Kaen University (IN62110). Piya Prajumwongs was supported by a scholarship from the Postgraduate Scholarship of Faculty of Medicine, Khon Kaen University. The funders had no role in study design, data collection and analysis, decision to publish, or preparation of the manuscript.

==============================
Reversine is a selective inhibitor of mitotic kinase monopolar spindle 1 (MPS1) and has been reported as an anticancer agent in various cancers. The effects of reversine on bile duct cancer, cholangiocarcinoma (CCA), a lethal cancer in Northeastern Thailand, were investigated. This study reports that reversine inhibited cell proliferation of CCA cell lines in dose- and time-dependent manners but had less inhibitory effect on an immortalized cholangiocyte cell line. Reversine also triggered apoptotic cell death by decreasing anti-apoptotic proteins, Bcl-XL and Mcl-1, increasing Bax pro-apoptotic protein and activating caspase-3 activity. Moreover, reversine induced autophagic cell death by increasing LC3-II and Beclin 1 while decreasing p62. Reversine activated autophagy via the AKT signaling pathway. Additionally, this study demonstrated for the first time that reversine could diminish the expression of Hypoxia-Inducible Factor 1- alpha (HIF-1α) and glucose transporter 1 (GLUT1), resulting in a reduction of glucose uptake and energy production in CCA cell lines. These findings suggest that reversine could be a good candidate as an alternative or supplementary drug for CCA treatment.

Introduction

Cholangiocarcinoma (CCA) is a cancer of bile duct epithelia. Problems of CCA are a high mortality rate related to late diagnoses and cancer metastasis. The highest incidence of CCA has been reported in Northeastern Thailand (Sripa et al., 2007; Sripa & Pairojkul, 2008). The liver fluke, Opisthorchis viverrini, is a major contributing cause of Thai CCA patients (Watanapa & Watanapa, 2002). 5-Fluorouracil (5-FU), the first line chemotherapy drug, is a choice of treatment for unresectable CCA patients, but it is not satisfactory. The outcome is often sub-optimal because CCA patients are commonly intrinsically resistant over time or will become refractory after initial 5-FU treatment, leading to CCA recurrence (Ramirez-Merino, Aix & Cortes-Funes, 2013). Therefore, the difficulty in these CCA patients is their poor response to the present chemotherapy; studies to identify new therapeutic drugs are therefore needed for therapy.

Reversine, a substituted purine derivative, is a selective inhibitor for monopolar spindle 1 (MPS1 or TTK), a mitotic checkpoint protein (Hiruma et al., 2016; Hiruma et al., 2017; Santaguida et al., 2010). This synthetic compound acts as an adenosine triphosphate (ATP) analogue which inhibits kinase enzyme activity (Hiruma et al., 2016).  Monopolar spindle protein 1 (MPS1) kinase has been reported to promote progression of cancers and MPS1 overexpression was related with poor survival of cancer patients including breast, liver, lung cancers (Choi et al., 2017; King et al., 2018; Tsai et al., 2020; Xu et al., 2016). Laboratory studies revealed that reversine has been shown to inhibit the growth of multiple cancers including colon (Park et al., 2019), prostate (Hsieh et al., 2007), oral squamous (Lee et al., 2012) and thyroid (Lu et al., 2012). The effects of reversine on cancer cells were G2/M cell cycle arrest, autophagy, and apoptosis in those cancers. Reversine also inhibited the phosphatidylinositol 3-kinase/protein kinase B (PI3K/AKT) signaling pathway which plays a role in biological processes including cell growth, anti-apoptosis and metabolism in cancer cells (Fresno Vara et al., 2004; Thorpe, Yuzugullu & Zhao, 2015). This inhibitory effect of PI3K/AKT signaling by reversine resulted in the inductions of autophagy and apoptosis (Lee et al., 2012; Kumar, Shankar & Srivastava, 2014; Shao et al., 2016). PI3K/AKT regulated GLUT1 and HIF-1α expressions; modulation of this signaling pathway led to cellular starvation including reduction of glucose uptake and ATP production (Melstrom et al., 2008; Zhang et al., 2016).

Upregulation of MPS1 was reported in OV-associated CCA tissues (Jinawath et al., 2006), therefore inhibition of MPS1 activity by reversine may block the G2/M phase of the cell cycle and induce cell death. Reversine may be an alternative choice for CCA treatment. This study aimed to investigate the effect of reversine on cell proliferation in CCA cells and related molecular mechanisms.

In the current study, the suppressive effects of reversine on viability of CCA cell lines were determined using the MTT assay, the cell cycle was analyzed by flow cytometry and the related molecular mechanisms of autophagy and apoptosis were investigated using Western blot analysis. The glucose uptake and ATP production were also measured. Results revealed that reversine induced G2/M arrest, apoptosis and autophagy via glucose and ATP deprivation in CCA cell lines.

Materials & Methods

Cell lines and reagent

CCA cell lines (KKU-100, KKU-213A and KKU-213B) were obtained from the Japanese Collection of Research Bioresources Cell Bank, Osaka, Japan and established as previously described (Sripa et al., 2005; Sripa et al., 2020). An immortalized cholangiocyte cell line, MMNK1 was previously characterized (Maruyama et al., 2004). Cells were cultured in high glucose (25 mM) Dulbecco’s Modified Eagle Medium (DMEM) supplemented with 10% heat inactivated fetal bovine serum (FBS) and a 1% antibiotic-antimycotic (Gibco, USA). Cells were incubated at 37 °C in a humidified 5% CO2 atmosphere. At 70–80% confluence, cells were detached from the culture flask using 0.25% w/v trypsin/EDTA and processed according to the particular assay. Reversine was purchased from Cayman Chemical (Michigan, USA) and prepared as a 100 mM stock in DMSO.

Cell proliferation assay

Cells were seeded in 96 well plates and treated with 0.1% DMSO as a vehicle control and various concentrations of reversine at 0.1, 0.5, 1, 10 and 20 µM, for 24, 48, and 72 h. MTT reagent was added into each well to a final concentration of 0.5 mg/ml and incubated for 4 h. DMSO was added to dissolve the insoluble formazan complex and the absorbance at 540 nm was measured. The data were analyzed at a half maximal inhibitory concentration (IC50) by the dose–response inhibition mode of the GraphPad Prism® 7.0 software (GraphPad software, Inc., San Diego, CA, USA).

Cell cycle analysis

KKU-213A and KKU-213B cell lines (2 × 105 cells) were seeded into a 6-well plate and treated with reversine (0, 1, 2 and 4 µM) for 24 h. Cells were collected, fixed with cold 70% ethanol overnight at 4 °C and treated with 0.1 mg/ml of RNase A at 37 °C for 1 h. Cells were stained with 10 µg/ml propidium iodide (PI) at 4 °C for 30 min in the dark. Cell cycle analysis was determined using a flow cytometer (BD LSR II™). FlowJo™ software was used to generate the acquisition and analysis plots of 20,000 cells.

Western blot analysis

CCA cells were lysed in buffer (50 mM Tris–HCl pH 7.5, 50 mM NaCl, 1%NP-40, 1% sodium deoxycholate) containing phosphatase and protease inhibitors. Protein lysates were applied on a 12% SDS-PAGE for Western blot analysis. Proteins were detected by antibodies including PI3K, AKT, phospho-AKT, GLUT1, LC3-II, Mcl-1, and cleaved caspase-3 (Cell Signaling Technology), HIF-1α (BD biosciences, USA), Bcl-XL, and Bax (Santa Cruz Biotechnology, Santa Cruz, CA, USA), Beclin 1 and p62 (Abcam, USA). GAPDH, (EMD Millipore, Germany) was used as an internal control. The immunoreactivity signals were captured and analyzed by ImageQuant™ TL analysis software (GE Healthcare, Chalfont, UK).

Glucose and ATP measurement

KKU-213A and KKU-213B cell lines (5 × 105 cells) were seeded onto 6-well plates and treated with a sublethal dose (IC25) at 4 µM, of reversine for 12 and 24 h. Glucose levels in the collected media were determined using a Glucose Colorimetric/Fluorometric Assay kit (K606, BioVision, USA) following the manufacturer’s protocol. Absorbance was measured at 570 nm and the glucose standard provided with the kit was used to calculate glucose levels. The glucose uptake was calculated by comparing the difference in leftover glucose concentrations between the vehicle and the reversine treatment. The amounts of ATP in cell lysates were measured and calculated from the standard curves using the ATP Colorimetric/Fluorometric Assay Kit according to the manufacturer’s protocol (K354; BioVision, Milpitas, CA, USA). The glucose uptake and ATP production of vehicle control were set as 100%.

Statistical analysis

Data were presented as mean ± SD. Statistical comparisons between groups were tested using the Student’s t-test. The data were analyzed by GraphPad Prism® 7.0 software (GraphPad software, Inc., San Diego, CA, USA) and SPSS 17.0 software (IL, USA). A P-value less than 0.05 was considered as statistically significant.

Results

Reversine inhibited cell proliferation in CCA cell lines

Three CCA cell lines, KKU-100, KKU-213A and KKU-213B, and an immortalized cholangiocyte cell line, MMNK1, were treated with various concentrations of reversine (0, 0.1, 0.5, 1, and 10 µM) for 24, 48 and 72 h. Cell viability was significantly decreased in dose- and time-dependent manners in all cell lines (* P < 0.05) (Figs. 1A–1C). The half-maximal inhibitory concentrations (IC50) of reversine at different time points for the 3 CCA cell lines were in the 0.62–10 µM range which were significantly lower than those of the MMNK1 cell line (7.05–20 µM range) as shown in Table 1. These results showed that reversine potentially reduced the cell viability of the CCA cell lines whereas it had less effect on the MMNK1 cell line. In addition, comparison between the cytotoxicity of reversine and 5-FU in CCA cell lines showed that the IC50 of reversine was lower than 5-FU at least 6–12 times (Fig. S1 and Table S1). This indicated that reversine was an effective drug and could be an alternative option for CCA treatment.

Reversine induced cell cycle arrest at the G2/M phase and apoptosis

The effects of reversine on the cell cycle were examined in KKU-213A and KKU-213B cell lines that were treated with increasing concentrations of reversine for 24 h and analyzed by flow cytometry (Fig. 2A). The results demonstrated that reversine significantly decreased the G1 phase and induced G2/M arrest in a dose-dependent manner. Reversine tended to increase the sub G1 population (apoptotic cells) in both cell lines as shown in Figs. 2B–2C. To confirm these phenomena, a Western blot assay was performed to check the protein expression levels related to G2/M arrest (cyclin B1, p21) and apoptosis (Bcl-XL and Mcl-1, Bax and cleaved caspase-3). KKU-213A and KKU-213B CCA cell lines were treated with 4 µM of reversine at different time points (12, 24, 48 and 72 h). The Western blot results showed a substantial decline of cyclin B1 while a significant rise in p21 in reversine treatment occurred. Furthermore, anti-apoptotic proteins including Bcl-XL and Mcl-1 were significantly decreased (* P < 0.05), whereas Bax pro-apoptotic protein and cleaved caspase-3 were increased noticeably in reversine treatment for 48 and 72 h in both CCA cell lines (Figs. 2D–2F).

Figure 1 Reversine inhibited cell proliferation of KKU-100, KKU-213A and KKU-213B CCA cell lines and had a minor effect on an immortalized cholangiocyte cell line, MMNK1.

Cells were treated with increasing concentrations (log10 scale) of reversine for (A) 24, (B) 48 and (C) 72 h. Cells treated with 0.1% DMSO were used as a vehicle control. The cell viability was determined using the MTT assay. Data are mean ± SD of three independent experiments. *P < 0.05 are shown in all cell lines when compared to vehicle.

Table 1 IC50 of reversine in three CCA cell lines, KKU-100, KKU-213A and KKU-213B and an immortalized cholangiocyte cell line, MMNK1.

Time (h)	IC50(µM)	
	MMNK1	KKU-100	KKU-213A	KKU-213B	
24	>20	5.51 ± 0.38	>10	>10	
48	16.71 ± 1.81	2.72 ± 0.32*	6.83 ± 2.61*	4.67 ± 0.77*	
72	7.05 ± 2.01	0.62 ± 0.42*	1.76 ± 0.94*	1.75 ± 0.82*	
Notes.

IC50 Half-maximal inhibitory concentration

* P-value < 0.05 compared with MMNK1 cell line at the same time points.

Figure 2 Reversine induced G2/M cell cycle arrest and apoptosis.

(A) Representative flow cytometry histograms of CCA cell lines treated with various concentrations of reversine for 24 h. The percentage of cells in each cell cycle phase are shown as mean ± SD from two independent experiments (B) KKU-213A and (C) KKU-213B. (D) Western blot analyses of the expression of cell cycles regulating proteins and apoptosis-related proteins upon reversine treatment. Cells were treated with 4 µM of reversine for 12, 24, 48 and 72 h. GAPDH was used as the internal control. The protein expression levels are reported as fold changes to vehicle control which was set as 1. (E) KKU-213A and (F) KKU-213B. Data shown are mean ± SD of three separate experiments. *P < 0.05 compared to vehicle control.

Reversine induced autophagy and reduced glucose uptake and ATP production via down-regulation of HIF-1α and GLUT1

The accumulation of cytoplasmic vacuoles that were similar to the morphological characteristics of autophagy were observed in reversine-treated KKU-213A and KKU-213B cell lines at 12 and 24 h (Fig. 3A). The expression of LC3-II, Beclin 1 and p62 proteins which are autophagy related markers, were examined in KKU-213A and KKU-213B cell lines treated with 4 µM of reversine for 12–72 h (Fig. 3B). A significant increase in LC3-II and Beclin 1 coupled with a decrease in p62 were shown in reversine treatment compared to vehicle controls (P < 0.05) (Figs. 3C–3D). These results suggested that reversine induced autophagy activation and autophagic flux occurred. To determine whether autophagic cell death induced by reversine treatment altered glucose metabolism, the glucose uptake and ATP production in reversine-treated CCA cells were examined. KKU213A and KKU213B CCA cells were treated with 4 µM reversine at 12 and 24 h. The cultured media were collected to measure the glucose uptake while the cell lysates were used to measure the ATP production. Reversine treatment significantly reduced glucose uptake at 24 h and reduced ATP production at 12 and 24 h as shown in Figs. 3E–3F. The protein expression of Hypoxia-inducible factor 1-alpha (HIF-1α) and glucose transporter 1 (GLUT1) by Western blot assay were further analyzed. The results showed markedly decreased protein expression of HIF-1α and GLUT1 in reversine treated cells compared with vehicle control (Figs. 3B–3D). Collectively, these results indicated that reversine treatment induced autophagic cell death and interfered with glucose uptake and ATP production via reduction of HIF-1α and GLUT1 protein expression levels.

Reversine induced autophagy via the AKT signaling pathway

The PI3K/AKT signaling pathway has been reported to regulate the autophagy. Therefore, the activation of PI3K/AKT after reversine treatment was determined by Western blot analysis. Figures 3B–3D demonstrated a significant decreased phosphorylation of the p110α catalytic subunit of PI3K (p110α) and the phosphorylation of serine 473 of AKT (s473) in reversine-treated CCA cell lines.

To further confirm that reversine induced autophagy via the AKT signaling pathway, SC-79, an AKT activator was used to activate the phosphorylation of AKT (s473). A combination of reversine and SC-79 recovered the cells from reversine-mediated autophagy (Figs. 4A–4B). Western blot results showed a tendency to increase pAKT, HIF-1α, GLUT1 and p62 expression with the concurrence of decreased Beclin 1 and LC3-II in reversine combined with SC-79 treatment (Fig. 4C). Among these, LC3-II was significantly decreased in the combination when compared to reversine treatment alone (Figs. 4D–4E, #P < 0.05).

Figure 3 Reversine induced autophagy via reduction of glucose uptake and ATP production.

(A) Cellular morphology of KKU-213A and KKU-213B cells after reversine treatment for 12 and 24 h were observed under phase contrast microscopy with 200× magnification. The red arrows indicate the cytoplasmic vacuoles. (B) Western blot was performed to detect the expression of PI3K/AKT signalling proteins, HIF-1α, GLUT1, Beclin 1, LC3II/I and p62. GAPDH detected in the same experiment as Fig. 2D was used as the internal control. The bar graphs represent the mean ± SD of expression levels (fold change to vehicle) from three independent experiments in (C) KKU-213A and (D) KKU-213B cell lines. (E) Glucose uptake and (F) ATP production were measured after reversine treatment at 12 and 24 h. Data are represented as mean ± SD of three independent experiments. P-values were calculated by Student’s t-test compared with vehicle (*P < 0.05).

Discussion

In this study, it was revealed for the first time that reversine could induce G2/M arrest, apoptosis and autophagic cell death in CCA cell lines. Reversine exhibited an anti-proliferative effect on all CCA cell lines tested in dose and time dependent manners. The IC50 of reversine at 72 h in Table 1 suggested that reversine had more effect (4 to 11-fold) on CCA cell lines than on the immortalized cholangiocyte. Noteworthy, KKU-100, a 5-FU resistant cell line was more sensitive to reversine than KKU-213A and KKU-213B (Table S1). This observation may be due to the alteration of some proteins after reversine treatment that may be involved in the cell viability of each cell line. Further investigation is required to determine the mechanisms which cause some chemo-resistant cell lines to be more sensitive than others to reversine. The above observation implies that reversine may be a potent anticancer drug for the chemotherapy resistant CCA cells. The effect and mechanism of reversine or in combination with a chemotherapy drug on chemotherapy resistant CCA cells should be interesting to explore. Furthermore, CCA cell lines were more sensitive to reversine than breast cancer cell lines (IC50 ∼ 5 µM) (Kuo et al., 2014) and cervical carcinoma cell lines (IC50 ∼ 7.5 µM) (Qin et al., 2013). This comparison hints that a low dose of reversine effectively inhibits the proliferation of CCA cells and may be a potential drug for CCA treatment or to augment current treatment. The antitumor activity of reversine in an animal model should be worth further investigation.

Figure 4 Reversine induced autophagy via suppression of AKT signaling.

Cellular morphology of (A) KKU-213A and (B) KKU-213B cells after reversine combined with SC-79 treatment for 24 h were observed under a phase contrast microscopy with 200× magnification. The red arrows indicate the cytoplasmic vacuoles. (C) Western blot was performed to detect the expression of pAKT/AKT, HIF-1α, GLUT1, Beclin 1, LC3II/I and p62. GAPDH was used as the internal control. The bar graphs represent the mean ± SD of expression levels (fold change to vehicle) from three independent experiments in (D) KKU-213A and (E) KKU-213B cells. *P < 0.05 compared to vehicle and #P < 0.05 compared to reversine alone.

The growth inhibitory effect of reversine on G2/M cell cycle arrest in CCA cell lines corresponded with the observations in several cancer cells after reversine treatment (Alves et al., 2016; D’Alise et al., 2008; Kuo et al., 2014; Lee et al., 2012; Lu et al., 2012). Western blot analysis supported the G2/M arrest in reversine-treated cells via the alteration of cyclin B1 and the cyclin-dependent kinase inhibitor (p21) which are the G2/M checkpoint markers. Similarly, decreased expression of cyclin B1 and increased expression of p21 during G2/M arrest after reversine treatment were also reported in prostate cancer (Hsieh et al., 2007) and renal cancer (Cheng et al., 2018).

Figure 5 The proposed molecular mechanisms of anti-tumor effects of reversine on CCA cells.

Reversine, an MPS1 inhibitor, inhibited cell proliferation of CCA cells by multiple mechanisms including G2/M arrest with changes of cyclin B1 and p21. This inhibitor caused autophagy activation and autophagic flux via alterations of autophagy related proteins. Additionally, reversine-mediated autophagy partially inactivated the AKT signalling pathway. Reversine also reduced the glucose uptake and ATP production via suppression of HIF-1α and GLUT1 expression. It consequently triggered apoptosis by decreasing of Bcl-XL and Mcl-1 expressions which subsequently inhibited Bax and activated the cleavage of caspase 3.

The present results showed reversine could induce apoptosis via down-regulation of anti-apoptotic proteins (Bcl-XL and Mcl-1), up-regulation of pro-apoptotic Bax protein and activation of caspase-3 in both CCA cell lines (Figs. 2D–2F). These results were consistent with the observations in oral squamous cell cancer (Lee et al., 2012), cervical carcinoma (Qin et al., 2013) and non-small cell lung cancer (Lu et al., 2016). The PI3K/AKT signaling pathway has been reported to regulate the apoptosis and autophagy (Baxt & Xavier, 2015; Duronio, 2008; Jee et al., 2002; Qian et al., 2009; Tsuruta, Masuyama & Gotoh, 2002; Wu et al., 2009). Therefore, the phosphorylation of PI3K (p110α) and AKT (s473) in reversine treated cells were examined. It was found that the PI3K/AKT signaling pathways were inactivated upon reversine treatment. This finding was similar to the studies in oral squamous cell (Lee et al., 2012) and thyroid cancers (Lu et al., 2012). Furthermore, reversine treatment led to the formation of cytoplasmic vacuoles and increased Beclin 1 and LC3-II, while it decreased p62 which indicated autophagy flux in both CCA cell lines. These findings were also found in the thyroid and non-small-cell-lung cancers (Lu et al., 2012; Lu et al., 2016).

Many studies have shown that PI3K/AKT signaling controlled glucose metabolism and energy production by regulating the expression of HIF-1α and GLUT1 (Hao, 2015; Katagiri et al., 1996). Whether reversine affected glucose uptake and ATP production in reversine-treated CCA cells was explored in the current study. It is reported for the first time that reversine treatment down-regulated HIF-1α and GLUT1 protein expression and consequently lowered glucose uptake and ATP production (Figs. 3B–3F). Consistently, these data were accompanied with the inactivation of PI3K/AKT signaling in reversine treatment (Figs. 4C–4E). Altogether, reversine disturbed the function of the MPS1 mitotic checkpoint protein and also interrupted PI3K/AKT signaling which is a major signal transduction pathway that involves cancer cell survival.

The overall anticancer effects and proposed molecular mechanisms of reversine on CCA cells are summarized in Fig. 5. Reversine, which is a selective inhibitor of MPS1, caused G2/M arrest with alteration of cyclin B1 and p21. It also induced apoptosis and autophagy. Various reports have shown that reversine can inhibit the in vitro kinase activity of aurora kinases, Janus Kinase 2 (JAK2), Rous Sarcoma (SRC) (McMillin et al., 2010) and ribosomal protein S6 kinase beta-1 (p70s6k) (Kim et al., 2007). Reversine may hamper the activity of JAK2 or SRC, consequently inhibiting the phosphorylation of PI3K/AKT signaling and resulting in the down-regulation of HIF-1α and GLUT1 expression. These phenomena diminish glucose uptake and ATP production and induce starvation-related autophagy. The current study showed that reversine-mediated autophagy inactivated AKT signaling. Moreover, down-regulation of HIF-1α by reversine suppresses the expression of transcription factors such as NF-kB, SP1 and CREB. This results in apoptosis induction through a decrease of Bcl-XL and Mcl-1 expressions which subsequently inhibit Bax and activated the cleavage of caspase 3.

Conclusions

Reversine markedly inhibited the cell proliferation of CCA cells by multiple mechanisms including G2/M cell cycle arrest, apoptosis and starvation-related autophagy. These new findings suggest that reversine could be a good candidate drug for CCA treatment. The effects of reversine itself or reversine in combination with current chemotherapy drugs are worthy of further preclinical and clinical investigations.

Supplemental Information

Figure S1 Dose response curve of 5-FU in CCA cell lines

CCA cells were treated with 0.25, 2.5, 25 and 250 µM of 5-FU for 48 h. Cells treated with 0.1% DMSO were used as a vehicle control. The cell viability was determined using the MTT assay. Data are mean ± SD of three independent experiments.

Click here for additional data file.

Table S1 IC50 of 5-FU in CCA cell lines

IC50: Half-maximal inhibitory concentration. * P-value < 0.05 compared with reversine.

Click here for additional data file.

Supplemental Information 3 MTT results 4 cell lines

Click here for additional data file.

Supplemental Information 4 Cell cycle analysis

Click here for additional data file.

Supplemental Information 5 Sub-G1 analysis

Click here for additional data file.

Supplemental Information 6 Original Western blot

Click here for additional data file.

Supplemental Information 7 Band intensities- Western blots

Click here for additional data file.

Supplemental Information 8 KKU213A

Click here for additional data file.

Supplemental Information 9 KKU213B

Click here for additional data file.

We would like to acknowledge Prof. James A. Will for editing the MS via Publication Clinic KKU, Thailand.

Additional Information and Declarations

Competing Interests

Author Contributions

Data Availability

The authors declare there are no competing interests.

Piya Prajumwongs performed the experiments, analyzed the data, prepared figures and/or tables, authored or reviewed drafts of the paper, and approved the final draft.

Orawan Waenphimai performed the experiments, analyzed the data, prepared figures and/or tables, and approved the final draft.

Kulthida Vaeteewoottacharn and Sopit Wongkham analyzed the data, authored or reviewed drafts of the paper, and approved the final draft.

Kanlayanee Sawanyawisuth conceived and designed the experiments, performed the experiments, analyzed the data, prepared figures and/or tables, authored or reviewed drafts of the paper, and approved the final draft.

The following information was supplied regarding data availability:

Original Western blot figures and band intensities are available in the Supplemental Files.

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
