# Peer review of "Reversine, a selective MPS1 inhibitor, induced autophagic cell death via diminished glucose uptake and ATP production in cholangiocarcinoma cells"

_PeerJ, doi:10.7717/peerj.10637_

## Round 0.1 · original submission · Major Revisions

Dear Authors,

Your manuscript has been read by three expert Reviewers. They all think that your work is important, however, they also present several critical comments that I ask you to very carefully address. Considering the need for additional animal experiments, proposed by Reviewer #3, I leave it to your consideration whether you add them or not. In case you successfully address all other concerns of the three Reviewers, your paper might be accepted even without the animal experiments.

·

Basic reporting

• The article lacks the use of clear and ambiguous English. There are several grammatical errors through the text. The current phrasing makes comprehension difficult.
• Background of disease and relevant use of reversine is not clear. Introduction fails to justify why the authors chose to test reversine in cholangiocarcinoma cell lines. Recent literature in the field is missing (for example, 2019 study in colorectal cancer https://www.spandidos-publications.com/10.3892/ijo.2019.4746) and there is no mention of cell cycle related changes in cholangiocarcinoma. Authors should consider explaining why is cholangiocarcinoma a candidate for reversine related G2/M arrest? There is no background on the relation of PI3K/Akt pathway, HIF1a or GLUT1 and autophagy in the introduction section.
• The structure of the article is well defined. The original western blot raw data .rar file is not accessible or corrupted.
• The authors defined a hypothesis but have not provided enough evidence to support the conclusions mentioned in the study.

---Custom checks – cell line checks – KKU-100, KKU-213A, KKU-213B and MMNK-1 are cholangiocarcinoma cell lines.

Experimental design

• The research is within the Aims and scope of the journal.
• The study identified the clear need for novel therapies in cholangiocarcinoma and has attempted to fill the gap. However, the authors failed to clearly justify the relevance of blocking G2/M phase of cell cycle and inducing autophagic cell death in cholangiocarcinoma. The authors should consider explaining the concept clearly.
• The investigation is conducted in a random manner. The experimental design is not clear. The method description for cell proliferation and cell cycle assays is not clear.
• In line 59, the cell lines are cultured in a DMEM media. Since the changes are associated with ATP differences in the untreated and treated cells, the author should explain the use of low-glucose or high-glucose DMEM.
• In line 106-110, the range of dosages tested is very few. Since the highest dose tested does not result in 100% death, the dose quoted as IC50 is not true IC50.In Figure 1, 48 hour graph fails to show 100% cell viability at the lowest dose, does this mean that the control or vehicle wells were different for all the time points? The method for drug dose response assay is not clear. In the method section, 5 dosages are tested, whereas in the figure authors are showing only 4 dose points? How was the background correction done? For MMNK-1, the graph shows a clear reduction of % viability. There is no p value or statistical evaluation of the results.
• In line 113-117, what is the ploidy level of these cell lines? How did they infer if the peaks of cell cycle are all from the same ploidy? Did they sort for nuclei? Why is G0-G1 phase not shown in the cell cycle plots? The authors should consider showing cell cycle differences at different time points. Depending on the doubling rate of both of these cell lines, the effect of a cell cycle inhibitor would be better understood in a time dependent manner. Is the G2/M arrest reversible or irreversible? A major predictor of G2-M checkpoint is CDK1 and cyclin A. Authors should consider showing other cell cycle checkpoint regulator proteins to validate the flow-cytometry based G2-M arrest.
• In figure 2C – for cleaved caspase 3 both the cell lines are showing different bands. Cleaved caspase 3 (CC-3) is depicted by a single and not an array of bands. Authors should comment on the CC-3 western blot image for KKU-213A. No statistics shown for western images.
• LC3BII/I blot is not clear. LC3BI band is not visible in the untreated cells, would this be linked to stress in the untreated cell lines? The authors should consider commenting on the autophagy state in the untreated cells and test other autophagy related markers like beclin1A or SQSTM1. The .rar file with all the original blots is not accessible.
• Authors have not tested the cell lines from a chemotherapy point of view so conclusions based on these speculations are too far-fetched. The authors have identified MMNK-1 as a normal cholangiocyte whereas it is an immortalized cholangiocyte cell line.

Validity of the findings

• The rationale and benefit to literature is not clearly stated. The statistical evaluation of cell proliferation and western blot images is missing. The conclusions made are not limited to supporting results. Several results are tied in with literature based speculations and too far-fetched.

Additional comments

Cholangiocarcinoma is a lethal disease and there is an urgent need for novel targeted therapies to improve patient outcome. I commend the authors for their attempt to propose a novel therapy for the treatment of cholangiocarcinoma. The overall approach used is clear and relevant however it lacks a well-defined thorough experimental analysis.
The data set is limited with the use of three cholangiocarcinoma cell lines and an immortalized cholangiocyte cell line. The cellular proliferation assays are not clearly defined and lack statistical power. Authors have used the proliferation assay data set to justify the use of chemotherapy-resistant cholangiocarcinoma; however, there is no evidence to support this hypothesis. Although the results are compelling, the cell cycle assays are limited to a single time point and do not display the sub G0 population or ploidy levels. Generally, these cell lines are slow growing and any cell cycle related changes should be analyzed based on their doubling rate. There is clearly an induction of apoptosis but it’s relation to autophagy is not clear. The most important issue is that the untreated cell line also has a LC3BII band but no LC3BI and a lack of statistical analysis. This could be improved by supplementing with other autophagy markers like beclin1A or SQSTM1.
Generally, the study is interesting and is worth publication but it needs to be majorly revised for clarity and statistical analysis.

Reviewer 2 ·

Basic reporting

The manuscript is generally clearly written and structured according to the standard format. The data in the figures and tables are well organized and presented.
However, there are a few points that could be improved:
1.The image for GAPDH is missing in Figure 3B, although it is mentioned in the figure legend.
2.Magnification is missing in Figure 3A. In addition, it would be easier for the reader to see the changes if images from higher magnification would be included.
3.The introduction could be expanded to give more information to the reader. The knowledge gap in the field could be further described. The topics studied in the manuscript, such as autophagy/ autophagic cell death, apoptosis, cell cycle arrest could be briefly introduced and the markers, which are being measured in this study, could be very shortly presented.

Experimental design

The research questions are well defined in the introduction and the experimental design of the study is generally good.
However, some points would need further clarification:
1.Glucose uptake assay: From the Methods section (lines 88-93), it is not very clear if glucose concentration in medium or glucose uptake was measured. There are 2 different kits provided by BioVision: Glucose Uptake Colorimetric Assay (catalogue number K676) and Glucose Colorimetric Assay (catalogue number K686), with which different parameters can be measured. Including the catalogue number of the kit and a brief description of how the assay was done, would clarify exactly which parameter was studied.
2.Figure legend for Figure 2: It is mentioned that the results shown in this figure were obtained from 2 independent experiments. For a better statistical significance of the results, at least one more independent experiment should be included.
3.Table 1: How the data on IC50 were calculated and obtained should be explained in detail in the Methods section.

Validity of the findings

The conclusions of the study are generally well stated.
There are a few points, which need to be addressed:
1.Figure 3B and line 127: An increase in the protein levels of LC3-II does not mean there is induction of autophagy. It means there is accumulation of LC3-II and autophagosomes, which could be caused BY increased production of LC3-II (as a consequence of induction of autophagy) OR BY inhibition of fusion autophagosome/autolysosome (which results in inhibition of autophagy). To differentiate between the 2 situations, measurement of autophagic flux could be performed (for example LC3-II Western blot in the presence of lysosomal inhibitors, such as bafilomycin A1; Klionsky D et al, Autophagy, 2016).
For clarification, the comments in the text could be changed or measurement of autophagic flux could be performed.
2.Results, Discussion, Figure 4: Throughout the text, there are statements on causal links, which are not supported by experimental results, such as: ‘Reversine induced autophagy and apoptosis via inhibiting PI3K/AKT signaling pathway’ (line 127). There are no data showing that the reduced PI3K/AKT signalling is the cause for reversine-autophagy induction. For this, analysis of the effect of modulation of PI3K/AKT on reversine-induced autophagy should be performed. This type of statements could be presented as opinions, speculations or hypothesis.

Additional comments

In this manuscript, the effects of reversine on different cholangiocarcinoma cell lines are investigated. Cell viability and markers of cell cycle, autophagy, apoptosis and insulin signaling are evaluated. The results indicate that reversine has an antiproliferative effect on these cancer cell lines. Thus, the findings of this study could be potentially interesting in the future for developing novel strategies for the treatment of cholangiocarcinoma.

Reviewer 3 ·

Basic reporting

The English language was clear unambiguous most of the time. Nevertheless, improvement is needed, especially in the introduction section. The introduction and background were very shallow scientifically. The authors talked about the unsatisfactory results of chemotherapy on unrespectable CCA. However, the authors did not detail, which chemotherapy specifically showed unsatisfactory results. The authors said the proposed treatment worked on multiple cancers but did not explain if these pieces of evidence were obtained from the laboratory or human studies (line 44-46). Also, the authors did not talk about the primary resistance mechanism. This is very important to show the gap in the knowledge in treating CCA. Also, this gap in the knowledge will put Reversine in the potential filling of this gap. As of now, the importance of Reversine in treating unrespectable CCA is not clear in comparison with the already available chemotherapies. Figures are relevant. However, some figures are not well labeled or not high quality.
In figure 2 A, color legends for the cell cycle phase were not clear. In figure 2 C, the quality of the western blot was not high. This was clear in multiple rows, including cleaved caspase-3, Bax, and Bcl-XL. In figure 3 A, the cell pictures were not clear: the type of scope used, the magnification have not been mentioned. The vacuoles of autophagy were not clear as well. In figure 3 B, the western blot's quality was not good as well, especially on the CLUT3. In addition to all of that, the authors did not talk about the current study's limitations.

Experimental design

The study is within the aim and the scope of this journal. However, the research question was not well defined. The authors did not identify the gap in knowledge and how this study research question will fill this gap. The investigation was acceptable with some significant concerns. Western plot experimental quality was weak and inconclusive. Some of the experiments have only repeated two times, which at least should be repeated in triplets. Although the authors used a negative control cell line of the immortal cholangiocytes, positive control was not used in this study using a resistant model to Reversine. Also, the authors used the vehicle but did not compare the standard treatment for CCA. Finally, this is just in vitro study of treatment of effect on cell survival, which is not enough and should be accompanied by in vivo treatment study to see if this treatment is working on these cell lines or patient-derived xenograft models on animals. The methodology could be improved to help in the reproducibility of these findings in other labs. For example, cell cycle analysis in lines 72-76 needs more details here and supplementary material, especially concerning the flow cytometry analysis and gating. Also, the authors decide to use four micromolar as the treatment dose for the subsequent experiments after the viability assays. However, it was not clear why this number. It was not clear why the authors continue the experiments only on two cell lines instead of all the three available.

Validity of the findings

The western plot quality was questionable and not helpful in drawing valid conclusions. The methodology was not novel. The conclusions are very firm for study depends only on in vitro results. Some of the studies' replicates need to be increased to at least three times. The authors found the treatment was effective against the chemo-resistant cell line (KKU-100). However, no clear explanations of why this happened and what is the possible explanation for such a phenomenon. The second sentence in the conclusion section (line 214-215) was written in a convincing tone that needs to be softened, especially that the authors did not take the opportunity to compare this proposed treatment to the standard chemotherapy for CCA, separately or in combination.

Additional comments

The study is a good start point to start building a substantial manuscript using more replicate, positive controls and additional experiments on animals. The authors should work more on the gab of the knowledge, its importance and how this new treatment will help in filling this gap. Also, authors should compare this treatment to the current standard chemotherapy for CCA.

---

## Round 0.2 · Minor Revisions

Dear authors,

Reviewers #1 and #2 have read the revised manuscript and are happy with the revisions. Reviewer #3 was no longer available, so the Academic Editor checked how you have addressed the comments of Reviewer #3.

The Academic Editor asks you to address this request experimentally: authors should compare reversine treatment to the current standard chemotherapy for CCA. Please include the additional experimental data, as well as a rebuttal letter, in your second revised version of the manuscript.

·

Basic reporting

There is a significant improvement in the manuscript. The text is clear and unambiguous. The background and introduction are well written.

Experimental design

The authors have addressed all previous concerns in the experimental design in the revised manuscript.

Validity of the findings

No comments.

Additional comments

The revised manuscript addresses all major concerns and is in a good shape to be published now.

Reviewer 2 ·

Basic reporting

No comment

Experimental design

No comment

Validity of the findings

No comment

Additional comments

The comments and questions were addressed and included in the manuscript.

---

## Round 0.3 · accepted · Accept

You have now addressed all remaining comments of the Reviewers.